# Impact of Extended and Restricted Antibiotic Deescalation on Mortality

**DOI:** 10.3390/antibiotics11010022

**Published:** 2021-12-27

**Authors:** Hwei Lin Teh, Sarimah Abdullah, Anis Kausar Ghazali, Rahela Ambaras Khan, Anitha Ramadas, Chee Loon Leong

**Affiliations:** 1Pharmacy Department, Hospital Kuala Lumpur, Ministry of Health Malaysia, Kuala Lumpur 50586, Malaysia; rahela.ak@gmail.com (R.A.K.); ramadas.anitha@gmail.com (A.R.); 2Biostatistics and Research Methodology Unit, Universiti Sains Malaysia (Health Campus), Kota Bharu 16150, Malaysia; sarimah@usm.my (S.A.); anisyo@usm.my (A.K.G.); 3Infectious Disease Unit, Hospital Kuala Lumpur, Ministry of Health Malaysia, Kuala Lumpur 50586, Malaysia; bkho@hotmail.com

**Keywords:** antibiotic, antimicrobial, de-escalation, streamlining, mortality, outcome, safety

## Abstract

Background: More data are needed about the safety of antibiotic de-escalation in specific clinical situations as a strategy to reduce exposure to broad-spectrum antibiotics. This study aims to compare the survival curve of patient de-escalated (early or late) against those not de-escalated on antibiotics, to determine the association of patient related, clinical related, and pressure sore/device related characteristics on all-cause 30-day mortality and determine the impact of early and late antibiotic de-escalation on 30-day all-cause mortality. Methods: This is a retrospective cohort study on patients in medical ward Hospital Kuala Lumpur, admitted between January 2016 and June 2019. A Kaplan–Meier survival curve and Fleming–Harrington test were used to compare the overall survival rates between early, late, and those not de-escalated on antibiotics while multivariable Cox proportional hazards regression was used to determine prognostic factors associated with mortality and the impact of de-escalation on 30-day all-cause mortality. Results: Overall mortality rates were not significantly different when patients were not de-escalated on extended or restricted antibiotics, compared to those de-escalated early or later (*p* = 0.760). Variables associated with 30-day all-cause mortality were a Sequential Organ Function Assessment (SOFA) score on the day of antimicrobial stewardship (AMS) intervention and Charlson’s comorbidity score (CCS). After controlling for confounders, early and late antibiotics were not associated with an increased risk of mortality. Conclusion: The results of this study reinforce that restricted or extended antibiotic de-escalation in patients does not significantly affect 30-day all-cause mortality compared to continuation with extended and restricted antibiotics.

## 1. Introduction

Antibiotic overconsumption and inappropriate antibiotic use remain the key drivers of bacterial resistance, with 30–50% of prescribed antibiotics being used inappropriately in hospital settings [1,2]. Antimicrobial resistance may result in clinical and economic adverse outcomes and a lack of new and effective antibiotics down the pipeline. Therefore, available broad-spectrum antibiotics must be used judiciously [3]. To address the increasing burden of multi-drug resistant bacterial infections, antimicrobial stewardship (AMS) programs are promoted to rationalize antibiotic prescription and conserve remaining antibiotics while improving patient outcomes. The current effort to improve antibiotic stewardship in Malaysia has been in its early stages since the national protocol on AMS was launched nationwide in 2014 [4]. The antimicrobial stewardship program strongly recommends de-escalation in order to promote judicious antimicrobial use and limit costs, adverse events, and the development of antibiotic resistance [5]. However, it is less commonly practiced than desired. Studies have shown that one of the main barriers is uncertainty regarding the safety of de-escalation, despite it being a standard of care among practicing physicians, especially in negative cultures [6,7]. Although the safety of de-escalation has been well established in various international studies, there is currently only one study in Malaysia on antibiotic de-escalation, which focuses on a single infection of ventilator-associated pneumonia in an intensive care unit [8]. Thus, offering more evidence for the safety of de-escalation will not only increase implementation, but also improve knowledge of the variables influencing the overall outcome of de-escalation. The aim of this study is (i) to compare the survival curves for de-escalation (early and late) and non-de-escalation on extended or restricted antibiotics; (ii) to determine the association of patient-related, clinically related, and pressure sore/device-related characteristics with the all-cause 30-day mortality of patients with suspected bacterial infection initiated on extended or restricted antibiotics; and (iii) to determine the impact of antibiotic de-escalation on all-cause 30-day mortality of patients with suspected bacterial infection initiated on extended and restricted antibiotics. We hypothesized that there would be no difference in survival probabilities between patients not de-escalated on antibiotics and those who had early or late de-escalation, while the prognostic factors for all-cause 30-day mortality of patients with suspected bacterial infection initiated with extended or restricted antibiotics would be patient-related, clinically related, and pressure sore/device-related characteristics. We also hypothesized that there would be no significant detrimental impact of early and late de-escalation on all-cause 30-day mortality.

## 2. Materials and Methods

### 2.1. Study Design

This was a retrospective cohort study on patients on extended and restricted antibiotics: Carbapenem (2016–2019) with the addition of patients on vancomycin and colistin (2018–2019) by reviewing medical record files in Hospital Kuala Lumpur. The accrual time for this study was three and half years, from 1 January 2016 to 30 June 2019, with an additional 1 month of follow up from 1 July 2019 to 31 July 2019.

### 2.2. Inclusion and Exclusion Criteria

Patients are included if they are aged ≥18 years old, admitted to the medical ward and started on Carbapenem (meropenem, imipenem/cilastatin, ertapenem), vancomycin, or colistin. Patients should also be reviewed by the antimicrobial stewardship team (AMS team) and deemed suitable for de-escalation. This AMS team, which consists of members recommended by Ministry of Health Malaysia (infectious disease physician, clinical pharmacists, clinical microbiologist, and an infection control nurse), will be prompted on cases initiated with Carbapenem, vancomycin, and colistin in medical wards. All such cases were reviewed Thursday of every week by the AMS team, and recommendation of de-escalation is communicated verbally directly to the primary treating team, who has the final decision on whether to accept the recommendation of de-escalation. Exclusion criteria of this study are those whose survival is less than 24 h after septic workup were drawn, if treatment was changed to another broader spectrum antibiotic (escalation of antibiotic), or if the patient was transferred in from another institution.

### 2.3. Data Collection

A data collection form was used to record all required information retrieved manually from patient’s medical file located in medical records. Such information included patient related characteristics, clinical related characteristics, pressure sore or device related characteristics, and if de-escalation has been performed.

### 2.4. Variables and Definition

The primary outcome of interest in this study was the event-death from all causes. Death status was verified by referring to death certificate in medical records retrieved from hospital archive center. The survival time was defined as the duration from the initiation of extended or restricted antibiotic to the date of the event. Patients still alive at study closure were censored on 31 July 2019. Antibiotic de-escalation is defined by changing an initially appropriate antimicrobial therapy from an empirical broad-spectrum characteristic to a narrower-spectrum one (by either changing the antimicrobial agent or by discontinuing an eventual antimicrobial combination, or both) according to culture results or clinical conditions, or shortening of the time course of the antimicrobial therapy, or withholding antibiotics. The classification and ranking of antibiotics was developed by consensus [9]. Early de-escalation was defined as de-escalation occurring within 4 days while late de-escalation was defined as de-escalation occurring beyond 4 days of extended or restricted antibiotic initiation. Censored was defined as alive or loss to follow up at day 30 days post antibiotic (extended or restricted antibiotic) initiation. Comorbidity defined as a pre-existing disease or condition in addition to the disease or condition designated as the principal diagnosis. The pre-existing disease had to be an active problem in one of two ways. Either the disease required treatment during the hospital admission, or the disease had permanently altered some organ function. Antimicrobial therapy administered before the susceptibility results were available was considered empirical. Therapy administered after microbiological report was considered microbiologically directed therapy. Indwelling catheter included temporary/permanent central venous device and percutaneous drainage. Source of infection in each patient was standardized according to Centre for Disease Control (CDC) criteria [10]. Sepsis severity was assessed using the Sequential Organ Failure Assessment (SOFA) [11,12]. Multidrug-resistant isolates were those producing Extended Spectrum β-lactamases (ESBLs) or *AmpC* β-lactamases, or Carbapenem-resistant. Investigations taken on Day 0 are investigations taken on the day of extended/restricted antibiotic initiation, or up to a maximum 48 h before initiation of extended/restricted antibiotic.

### 2.5. Statistical Analysis

Survival analysis was carried out by Kaplan–Meier survival curves and analysed by the Fleming–Harrington test for the first objective. Besides the Kaplan–Meier survival curve, simple univariable and multivariable Cox regression analysis was performed for the second and third objective. The regression coefficient (b) with standard error (SE), adjusted hazard ratio (AHR) with its 95% confidence interval, Wald statistics and its corresponding *p* value were reported. Variables with *p* value < 0.25 were selected to be included in multivariable analysis. Methods used for the selection of variables to be included in the model are forced entry, forward stepwise, and backward stepwise. In this process, the probability of entry (Pe) and the probability of removal (Pr) are pre-determined as 0.05 and 0.1, respectively, throughout the whole variable selection process. The preliminary final model was checked for multicollinearity, specification error, and proportionality of hazard assumption. Data analysis was performed using STATA SE Version 14. The sample size required for this study to have an 80% power to detect a 70% difference survival time of de-escalated vs. non-de-escalated group with a two-sided test with an a level of 0.05 was calculated to be 172. The 70% difference in survival time was based on expert opinion of an infectious disease consultant as previous studies on the safety of de-escalation were largely undertaken with logistic statistical analysis and no data on difference in median survival time were readily available. Sample size was calculated using power and sample size calculation (PS) Software.

## 3. Results

A total of 180 subjects fulfilled the inclusion criteria, and because the sample size calculated approximates sampling frame no probability sampling was applied in this study. All 180 subjects were included in the final analysis, and all subjects completed follow-up. Overall, there were 62 deaths (34.4%) and 118 censored events (65.6%). The 118 subjects were censored because death did not occur at the end of follow-up. Out of 180 patients seen by the AMS team, 132 (73.3%) cases were successfully de-escalated on extended or restricted antibiotics, of which 79 patients (43.9%) had early de-escalation while 53 patients (29.4%) had late de-escalation. The main de-escalation was discontinuation of extended and restricted antibiotic (37.8%), followed by changing to a narrow spectrum antibiotic (31.7%) and shortening of the duration of antibiotic therapy (3.8%). Patient characteristics, clinical characteristics, pressure sore or device related characteristics between the groups of de-escalation are shown in Table 1, Table 2 and Table 3. Simple and multiple survival regression analyses of patient, clinical, and pressure sore or devise related variables are shown in Table 4, Table 5, Table 6 and Table 7.

### 3.1. Survival Curve of Those De-Escalated and Non-De-Escalated on Antibiotics

In Fleming-Harrington test, the overall mortality rates were not significantly different when patient was not de-escalated on extended or restricted antibiotics, to those de-escalated early or later (*p* = 0.760). Figure 1 show graphical illustration of Kaplan-Meier survival curve between the de-escalation group.

### 3.2. Variables Associated with All Cause 30-Days Mortality

The univariable analysis of variables associated with all cause 30-days mortality is outlined in Table 4, Table 5, Table 6 and Table 7. Multivariable analysis associated with all cause 30-days mortality were Sequential Organ Function Assessment (SOFA) score on the day of antimicrobial stewardship (AMS) intervention (AHR 6.61, 95% CI 3.90,11.18; *p* < 0.001) and Charlson’s comorbidity score (AHR 1.97, 95% CI 1.17,3.30; *p* = 0.01). Multicollinearity and interactions were not observed. The preliminary final model was properly specified (Table 8). Hazard function plots, Log-minus-log plots, Schoenfeld partial residual plots, as well as scaled and non-scaled Schoenfeld residuals test indicated proportionality of hazard. Regression diagnostics were performed by Cox–Snell residual analysis, which indicated that the model is a good fit, while Harrell’s C statistic was calculated to assess the discrimination ability of the preliminary final model. The C-statistics was 0.795, suggesting acceptable discrimination.

Forward, backward, and stepwise Cox proportional hazards regression model applied.Multicollinearity and interactions were not observed.The preliminary final model was properly specified.Hazard function plots, Log-minus-log plots, Schoenfeld partial residual plots, scaled and non-scaled Schoenfeld residuals test, and C-statistics were applied to check the assumption of the model.Regression diagnostics were performed by Cox–Snell residual, Martingale residual, deviance residual, and influential analysis.Influential outliers were identified by checking percent changes in regression coefficient.

### 3.3. Impact of De-Escalation on 30-Day All-Cause Mortality

Forced entry of AMS de-escalation into the final model (Table 9) indicated that patients de-escalated on extended or restricted antibiotics, whether early or late de-escalation, did not have a detrimental impact on 30-day all-cause mortality compared to continuation with extended and restricted antibiotics, after adjusting for confounders. The AHR for early and late de-escalation was 0.67 (95% CI 0.36,1.22, *p* = 0.194) and 0.70 (95% CI 0.35,1.41; *p* = 0.321) respectively.

## 4. Discussion

The overall rate of de-escalation in this study was 73%. Several recent studies on de-escalation, most of which included mostly patients with identified pathogens, have documented de-escalation rates of 23–68% [13,14,15,16,17,18,19]. In contrast to the aforementioned studies, the current study included patients with and without a microbiological diagnosis. Therefore, the de-escalation rate achieved (73%) was slightly higher than those described in previous studies [13,20,21,22]. The higher overall rate of de-escalation, despite the absence of a microbiological diagnosis, may be explained by the presence of an AMS team in the hospital, who were available to prompt and recommend de-escalation to the primary team. The rate of de-escalation was also higher than that of intensive care unit (ICU) settings, as the current study involved less critically ill patients [8]. Early de-escalation in this study was slightly lower than that in the study by performed by Liu et al. [23], despite a similar hospital setting and the presence of an AMS team, due to varying definitions of de-escalation. When compared to studies with the same definition of de-escalation, the rates of early de-escalation and late de-escalation were similar to those in a study by Palacios-Baena, et al. [24]. The frequency of normal WBC counts (<×10^9^/L) on day 0 and intervention was significantly higher in non-deescalated group than in early- and late-deescalated groups. This was because the primary team generally refuse to de-escalate once a patient has been shown to respond to an antibiotic regimen, as shown in normalization of white cell count, and would tend to continue and complete the antibiotic regimen.

Fleming–Harrington tests comparing overall mortality rates showed no significant difference between patients not de-escalated on extended or restricted antibiotics and those de-escalated early or later (*p* = 0.760). This result was in concordance with a study involving similar hospital settings conducted by Koupetori, et al. [25] on the survival of patients with bloodstream infection, in which log-rank test results were reported to be *p* = 0.683. Two other studies conducted in ICU settings also generated similar results [8,26].

Two variables were found to be associated with all-cause 30-day mortality of patients initiated with extended or restricted antibiotics: Charlson’s comorbidity score (CCS) and SOFA score on AMS intervention day. The strength of the association for CCS was documented by Palacious-Baena et al. [24] with slight differences in AHR attributed to two reasons: first, the difference in CCS cut-offs, which is lower in our current study, and second, the difference in population diagnosis, for which Palacious-Baena et al. included only confirmed blood infection patients, while the current study included heterogeneous infection cases. SOFA scores >4 on the day of AMS intervention were also highly associated with mortality, reinforcing findings previously described in the literature, in which patient severity was an important factor in establishing prognosis after infection [27,28,29,30]. The strength of association was much higher compared to that in a study including only gram-negative bloodstream infection, in which it was found that patients with SOFA scores >4 have only double the risk of 30-day mortality with HR 2.18 (95% CI 1.03, 4.57; *p* = 0.03) [21]. Such a discrepancy may occur because SOFA scores were recorded at different time.

The overall mortality rates were not significantly different when patients were not de-escalated after controlling for confounders. The results from the current study are in accordance with findings from a prospective, multicenter cohort conducted by Palacious-Baena et al. The close similarity to the current study can most likely be attributed to several factors. First, the cut-off points for early and late de-escalation were similar for both studies. They also involved similar groups of antibiotics from which patients were de-escalated, namely imipenem, ertapenem, and meropenem. Third, the other study controlled for CCS and SOFA scores, which were also found to be associated with mortality in the current study. Similar results were again reiterated by Koupetori et al. [25] after controlling for confounders of septic shock/sepsis, age, gender, and concomitant disease.

To the best of our knowledge, this is the first Malaysian study that has focused on the impact of AMS de-escalation on patients using extended and restricted antibiotics. The interventions recommended by the AMS team did not compromise patient clinical outcome, which in this study is all-cause 30-day mortality.

This study has several strengths. First, it was conducted at the largest tertiary hospital under the Ministry of Health of Malaysia; hence, the results can be applied to other Malaysian hospitals with an AMS team. The mortality data were also derived from reliable documentation: the death registry and death certificate issued by Hospital Kuala Lumpur. Since the study had an objective clinical outcome that could be tracked, bias was not possible. One disadvantage of mortality as a clinical outcome is that if there are any changes in mortality, it is difficult to attribute those changes directly to an intervention. Hence, this study attempted to adjust for several confounders that could affect mortality, such as underlying comorbidities and severity of infection.

There are several limitations inherent to the design of this study. The retrospective design of our study is a methodological limitation, which is difficult to overcome because of the obvious ethical issues that must be considered when studying the management of a life-threatening illness. Analyzing patients in our cohort retrospectively may have resulted in the possibility of information bias and limited ability to study barriers to the de-escalation of extended and restricted antibiotics. Secondly, it is difficult to distinguish true pathogens from colonization. The suspicion of infection and the decision to obtain cultures, septic workup, or the choice and doses of antimicrobials depended mostly on the primary care physicians rather than being guided by a protocol or recommendations made by infectious disease specialists. In addition, this study assessed comorbidities retrospectively, which can result in an underestimation of their true prevalence. This study also involved a heterogeneous mix of infections and organisms that were included and analyzed collectively. Infections by these bacteria may potentially carry different risk factors and prognoses. The association between different mechanisms of resistance (e.g., *AmpC*, Extended beta-lactamases, Carbapenemese resistance) and outcomes of infection remains unclear in this study. Hence, it remains uncertain whether a de-escalation strategy can be implemented for infections caused by other, potentially antibiotic-resistant pathogens. Despite these shortcomings, we believe that the findings in this study establish several clinical variables that can help clinicians to identify patients at high risk of mortality.

This study has shown there is no difference in overall mortality if a patient is de-escalated on extended or restricted antibiotics in medical wards. An association was found between the CCS and SOFA scores. These interesting observations could lead to further studies being conducted to understand the basis for these differences. Although CCS and SOFA scores are unmodifiable factors, understanding these differences and risk factors is important in the development of prediction models and personalized treatment. Practitioners can utilize such scores as a guide in the escalation of supportive therapy and other interventions, such as infection source control. If necessary, the family members of such groups can be informed regarding the chances of death during end-of-life or palliative care counseling.

For future studies, a bigger sample size is necessary so stratification according to the cause of death, either infection related or non-infection related, can be performed in addition to all-cause mortality. Since the current study is retrospective, the classification of mortality being either infection or non-infection related is difficult and can be biased; hence, a prospective study may overcome such a limitation. Despite limited evidence supporting the validity of mortality as a measure of stewardship programs, it remains an important patient-centered outcome. Future studies may consider investigating other clinically relevant outcomes, such as hospital readmission rates due to infection or recurrent infections, while disease-related mortality should primarily be used as a secondary or exploratory outcome. All-cause mortality can be reported in addition to disease-related mortality.

## 5. Conclusions

This study reinforces the fact that restricted or extended antibiotic de-escalation in patients does not have a detrimental impact on all-cause 30-day mortality compared to patients continued with restricted or extended antibiotics.

## Figures and Tables

**Figure 1 antibiotics-11-00022-f001:**
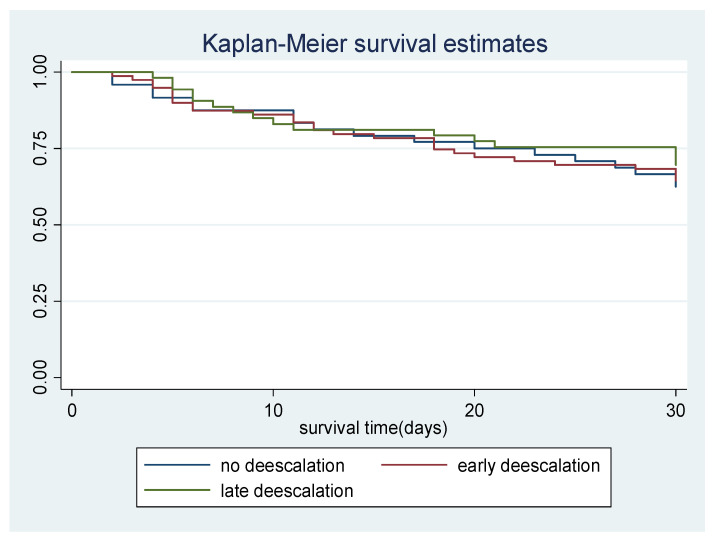
Kaplan-Meier estimates for overall survival rates based on de-escalation group; no de- escalation, early de-escalation and late de-escalation.

**Table 1 antibiotics-11-00022-t001:** Frequency distribution of patient related characteristics based on de-escalation group.

Patient Related Characteristics	No De-Escalation *n* = 48 Frequency (%)	Early De-Escalation *n* = 79 Frequency (%)	Late De-Escalation *n* = 53 Frequency (%)	*p*-Value
Age				
Age ≤ 65 years	30 (26.6)	50 (44.2)	33 (29.2)	0.992
Age > 65 years	18 (26.9)	29 (43.3)	20 (29.8)	
Gender				
Male	20 (22.0)	40 (44.0)	31 (34.0)	0.240
Female	28 (31.5)	39 (43.8)	22 (24.7)	
Ethnicity				
Malay	22 (23.7)	41 (44.1)	30 (32.2)	0.464
Chinese	12 (30.0)	14 (35.0)	14 (35.0)	
Indian	9 (25.7)	19 (54.3)	7 (20.0)	
Others	5 (42.0)	5 (42.0)	2 (16.0)	
ICU Stay				
No	39 (27.7)	66 (46.8)	36 (25.5)	0.087
Yes	9 (23.1)	13 (33.3)	17 (43.6)	
Invasive Mechanical Ventilation				
No	32 (29.1)	51 (46.4)	27 (24.5)	0.190
Yes	16 (22.9)	28 (40.0)	26 (37.1)	
CCS				
0–2	30 (27.3)	45 (40.9)	35 (31.8)	0.562
≥3	18 (25.7)	34 (48.6)	18 (25.7)	
McCabe Score				
1	41 (25.2)	70 (42.9)	52 (31.9)	0.068
≥2	7 (41.2)	9 (52.9)	1 (5.9)	
Illicit drug use				
No	48 (27.3)	76 (43.2)	52 (29.5)	0.364
Yes	0 (0.00)	3 (75.0)	1 (25.0)	
Smoking status				
Non smoker	37 (28.2)	56 (42.8)	38 (29.0)	0.928
Ex-smoker	4 (19.1)	10 (47.6)	7 (33.3)	
Active smoker	7 (25.0)	13 (46.4)	8 (28.6)	
History of Hospital Admission within 3 months				
No	33 (27.1)	53 (43.3)	36 (29.5)	0.981
Yes	15 (25.9)	26 (44.8)	17 (29.3)	
History of antibiotic exposure within 3 months				
No	38 (29.7)	50 (39.0)	40 (31.2)	0.113
Yes	10 (19.2)	29 (55.8)	13 (25.0)	
Presence of ESRF				
No	47 (27.3)	75 (43.6)	50 (29.1)	0.642
Yes	1 (12.5)	4 (50.0)	3 (37.5)	
Diabetes with end organ failure				
No	35 (26.5)	59 (44.7)	38 (28.8)	0.928
Yes	13 (27.1)	20 (41.7)	15 (31.2)	
Presence of HIV				
No	48 (27.1)	76 (42.9)	53 (30.0)	0.142
Yes	0 (0.00)	3 (100.0)	0 (0.00)	
Presence of Malignancy				
No	45 (27.8)	69 (42.6)	48 (29.6)	0.499
Yes	3 (16.7)	10 (55.6)	5 (27.8)	

**Table 2 antibiotics-11-00022-t002:** Frequency distribution of clinical related characteristics based on de-escalation group.

Clinical Related Characteristics	No De-Escalation *n* = 48 Frequency (%)	Early De-Escalation *n* = 79 Frequency (%)	Late De-Escalation *n* = 53 Frequency (%)	*p*-Value
Acquisition of infection				
Community acquired	26 (26.3)	42 (42.4)	31 (31.3)	0.826
Hospital or healthcare	22 (27.2)	37 (45.7)	22 (27.1)	
acquired				
Extended or Restricted antibiotic initiated				
Meropenem	36 (29.5)	47 (38.5)	39 (32.0)	0.672
Imipenem	4 (16.0)	14 (56.0)	7 (28.0)	
Ertapenem	6 (23.1)	14 (53.9)	6 (23.0)	
Colistin	1 (25.0)	2 (50.0)	1 (25.0)	
Vancomycin	1 (33.3)	2 (66.7)	0 (0.00)	
Therapy of antibiotic				
Empirical	35 (30.7)	45 (39.5)	34 (29.8)	0.193
Microbiologically	13 (19.7)	34 (51.5)	19 (28.8)	
directed				
Source of infection				
Others	32 (32.0)	38 (38.0)	30 (30.0)	0.122
Respiratory	16 (20.0)	41 (51.2)	23 (28.8)	
Aetiology (sterile culture)				
No growth	41 (28.9)	59 (41.5)	42 (29.6)	0.658
Others	4 (19.1)	12 (57.1)	5 (23.8)	
*Klebsiella pneumonia*	3 (23.0)	5 (38.5)	5 (38.5)	
Resistance (sterile culture)				
Sensitive strain or Others	3 (23.1)	9 (69.2)	1 (7.7)	0.082
Multidrug resistant	4 (19.1)	8 (38.0)	9 (42.9)	
isolate ^a^				
CRP (mg/L) on Day 0 ^b^				
25–64	10 (32.3)	13 (41.9)	8 (25.8)	0.662
65–143	5 (20.0)	9 (36.0)	11 (44.0)	
144–240	2 (13.3)	7 (46.7)	6 (40.0)	
>240	3 (33.3)	5 (41.7)	3 (25.0)	
Temperature (°C) on Day 0				
≤37.5	23 (31.5)	28 (38.4)	22 (30.1)	0.376
>37.5	25 (23.4)	51 (47.6)	31 (29.0)	
White cell count (×109/L) on Day 0				
≤11	28 (40.6)	26 (37.7)	15 (21.7)	0.003 *
>12	20 (18.0)	53 (47.8)	38 (32.2)	
Platelet (×103 /μL) Day 0				
≥150	12 (33.3)	16 (44.5)	8 (22.2)	0.461
<150	36 (25.0)	63 (43.8)	45 (31.2)	
SOFA score Day 0				
≤4	37 (29.1)	55 (43.3)	35 (27.6)	0.463
>4	11 (20.8)	24 (45.3)	18 (33.9)	
Severity of infection Day 0				
Not in sepsis	29 (29.0)	41 (41.0)	30 (30.0)	0.724
Sepsis	12 (21.4)	26 (46.5)	18 (32.1)	
Septic shock	7 (29.2)	12 (50.0)	5 (20.8)	
Albumin level Day 0 (g/L) ^c^				
Mild hypoalbuminemia (25–35)	12 (25.5)	22 (46.8)	13 (27.7)	0.709
Severe hypoalbuminemia (<25)	34 (26.7)	56 (44.1)	37 (29.1)	
CRP/Albumin ratio Day 0 ^d^				
≤2	8 (33.4)	11 (45.8)	5 (20.8)	0.508
>2	11 (20.8)	22 (41.5)	20 (37.7)	
White cell count (×109/L) on intervention day				
≤11	32 (35.6)	36 (40.0)	22 (24.4)	0.024 *
>12	16 (17.8)	43 (47.8)	31 (34.4)	
SOFA score on intervention day				
≤4	37 (27.8)	56 (42.1)	40 (30.1)	0.708
>4	11 (23.4)	23 (48.9)	13 (27.7)	
Severity of infection on intervention day				
Not in sepsis	37 (31.4)	47 (39.8)	24 (28.8)	0.092
Sepsis	6 (12.7)	24 (51.1)	17 (37.2)	
Septic shock	5 (33.3)	8 (53.3)	2 (13.4)	

^a^ Multidrug-resistant isolates were those producing ESBLs or *AmpC*, or carbapenem-resistant; ^b^ 53.9% missing values (*n* = 97), ^c^ 3.3% missing value (*n* = 6); ^d^ 57.2% missing values (*n* = 103). * Statistically significant difference was found between no de-escalation vs early de-escalation, and no de-escalation vs late de-escalation.

**Table 3 antibiotics-11-00022-t003:** Frequency distribution of pressure sore and device related characteristics based on de-escalation group.

Pressure Sore and Device Related Characteristics	No De-Escalation *n* = 48 Frequency (%)	Early De-Escalation *n* = 79 Frequency (%)	Late De-Escalation *n* = 53 Frequency (%)	*p*-Value
Presence of indwelling CVC				
No	33 (27.1)	52 (42.6)	37 (30.3)	0.878
Yes	15 (25.8)	27 (46.6)	16 (27.6)	
Presence of indwelling urinary catheter				
No	26 (29.2)	39 (43.8)	24 (27.0)	0.672
Yes	22 (24.2)	40 (44.0)	29 (31.9)	
Presence of pressure sore				
No	37 (31.6)	49 (41.9)	31 (26.5)	0.120
Yes	11 (17.7)	29 (46.8)	22 (35.5)	

**Table 4 antibiotics-11-00022-t004:** Patient related factor of all-cause 30-days mortality in patients suspected with bacterial infection on extended or restricted antibiotic using simple Cox proportional hazards regression model (*n* = 180).

Variables	Event *n* = 62, Frequency (%)	Censored *n* = 118, Frequency (%)	b (SE)	Crude Hazards Ratio (95% CI)	Wald Statistic	*p*-Value
Age						
Age ≤ 65 years	36 (58.1)	77 (65.3)	0	1		
Age > 65 years	26 (41.9)	41 (34.7)	0.20 (0.26)	1.22 (0.74–2.03)	0.77	0.441
Gender						
Male	30 (48.4)	61 (51.7)	0	1		
Female	32 (51.6)	57 (48.3)	0.05 (0.25)	1.06 (0.64–1.74)	0.21	0.832
Ethnicity						
Malay	27 (43.6)	66 (55.9)	0	1		
Chinese	15 (24.2)	25 (21.2)	0.48 (0.32)	1.62 (0.85–3.08)	1.47	0.142
Indian	15 (24.2)	20 (16.7)	0.51 (0.33)	1.67 (0.88–3.17)	1.57	0.117
Others	5 (8.0)	7 (6.2)	0.52 (0.49)	1.68 (0.64–4.40)	1.06	0.289
ICU Stay						
No	11 (17.7)	29 (24.6)	0	1		
Yes	51 (82.3)	89 (75.4)	−0.39 (0.35)	0.68 (0.34–1.34)	0.26	0.264
Invasive Mechanical Ventilation						
No	34 (54.8)	36 (30.5)	0	1		
Yes	28 (45.2)	82 (69.5)	0.87 (0.26)	2.38 (1.43–3.94)	3.35	0.001
CCS						
0–2	28 (45.2)	82 (69.5)		1		
≥3	34 (54.8)	36 (30.5)	0.84 (0.26)	2.32 (1.40–3.86)	3.26	0.001
McCabe Score						
1	51 (82.3)	112 (94.9)	0	1		
≥2	11 (17.7)	6 (5.1)	0.81 (0.34)	2.26 (1.17–4.37)	2.41	0.016
Illicit drug use						
No	59 (95.2)	117 (99.2)	0	1		
Yes	3 (4.8)	1 (0.8)	1.11 (0.59)	3.03 (0.95–9.73)	1.87	0.062
Smoking status						
Non smoker	44 (71.0)	87 (73.7)	0	1		
Ex-smoker	7 (11.3)	14 (11.9)	0.14 (0.41)	1.15 (0.51–2.56)	0.34	0.737
Active smoker	11 (17.7)	17 (14.4)	0.18 (0.34)	1.20 (0.62–2.33)	0.54	0.590
History of Hospital Admission within 3 months						
No	44 (71.0)	78 (66.1)	0	1		
Yes	18 (29.0)	40 (33.9)	−0.17 (0.28)	0.84 (0.49–1.46)	−0.60	0.547
History of antibiotic exposure within 3 months						
No	46 (74.2)	82 (69.5)	0	1		
Yes	16 (25.8)	36 (30.5)	−0.16 (0.29)	0.85 (0.48–1.50)	−0.56	0.575
Presence of ESRF						
No	57 (91.9)	115 (97.5)	0	1		
Yes	5 (8.1)	3 (2.54)	0.87 (0.47)	2.38 (0.95–5.97)	1.86	0.063
Diabetes with end organ failure						
No	39 (63.9)	93 (78.8)	0	1		
Yes	23 (37.1)	25 (21.2)	0.60 (0.27)	1.82 (1.08–3.06)	2.26	0.024
Presence of HIV						
No	59 (95.2)	118 (100.0)	0	1		
Yes	3 (4.8)	0 (0.0)	1.46 (0.60)	4.31 (1.34–13.81)	2.46	0.014
Presence of Malignancy						
No	52 (83.9)	110 (93.2)	0	1		
Yes	10 (16.1)	8 (6.8)	0.89 (0.35)	2.42 (1.22–4.80)	2.55	0.011
Chronic liver failure						
No	54 (87.1)	114 (96.6)	0	1		
Yes	8 (12.9)	4 (3.4)	1.03	2.80 (1.33–5.90)	2.70	0.007

**Table 5 antibiotics-11-00022-t005:** Clinical related factor of all-cause 30-days mortality in patients suspected with bacterial infection on extended or restricted antibiotic using simple Cox proportional hazards regression model (*n* = 180).

Variables	Event *n* = 62 Median (IQR)/ Frequency (%)	Censored *n* = 118 Median (IQR)/ Frequency (%)	b (SE)	Crude Hazards Ratio (95% CI)	Wald Statistic	*p*-Value
Acquisition of infection						
Community acquired	25 (40.3)	74 (62.7)	0	1		
Hospital or healthcare	37 (59.7)	44 (37.3)	0.60 (0.26)	1.83 (1.09–3.06)	2.30	0.022
acquired						
Therapy of antibiotic						
Empirical	47 (75.8)	67 (56.8)	0	1		
Microbiologically	15 (24.2)	51 (43.2)	−0.64 (0.30)	0.53 (0.29–0.95)	−2.14	0.033
directed						
Duration of Extended or Restricted antibiotic	4 (4) *	5 (5) *	−0.05 (0.04)	0.95 (0.88–1.02)	−1.49	0.137
Source of infection						
Non respiratory	27 (43.6)	73 (61.9)	0	1		
Respiratory	35 (56.4)	45 (38.1)	0.56 (0.26)	1.76 (1.06–2.91)	2.20	0.028
Aetiology (sterile culture)						
No growth	51 (82.2)	91 (79.8)	0	1		
Others	7 (11.3)	18 (12.3)	−0.01 (0.40)	0.99 (0.45–2.19)	−0.02	0.981
*Klebsiella Pneumonia*	4 (6.5)	9 (7.9)	−0.29 (0.52)	0.97 (0.35–2.70)	−0.06	0.955
Resistance (sterile culture)						
Sensitive strain or Others	4 (36.4)	9 (39.1)	0	1		
Multidrug resistant isolate ^a^	7 (63.6)	14 (60.9)	0.10 (0.63)	1.11 (0.32–3.78)	0.87	0.872
CRP (mg/L) on Day 0 ^b^						
25–64	8 (40.0)	23 (36.5)	0	1		
65–143	5 (25.0)	20 (31.7)	−0.09	0.91 (0.29–2.88)	−0.15	0.878
144–240	5 (25.0)	10 (15.9)	0.45	1.58 (0.50–4.98)	0.78	0.434
>240	2 (10.0)	10 (15.9)	−0.32	0.73 (0.15–3.51)	−0.40	0.693
Temperature(°C) on Day 0						
≤37.5	22 (35.5)	51 (43.2)	0	1		
>37.5	40 (64.5)	67 (56.8)	0.30 (0.27)	1.35 (0.80–2.27)	1.13	0.257
White cell count (× 109/L) on Day 0						
≤11	28 (45.2)	41 (34.8)	0	1		
>12	34 (54.8)	77 (65.2)	−0.22 (0.26)	0.80 (0.48–1.32)	−0.87	0.383
Platelet (×103 /μL) Day 0						
≥150	22 (35.5)	14 (11.9)	0	1		
<150	40 (64.5)	104 (88.1)	−0.88 (0.27)	0.41 (0.25–0.70)	−3.30	0.001
Albumin level Day 0 (g/L) ^c^						
Mild hypoalbuminemia						
(25–35)	10 (16.1)	37 (33.0)	0	1	2.45	0.014
Severe hypoalbuminemia						
(<25)	52 (83.9)	75 (67.0)	0.89 (0.36)	2.43 (1.20–4.93)		
SOFA score Day 0						
≤4	26 (42.0)	101 (85.6)	0	1		
>4	36 (58.0)	17 (14.4)	1.63 (0.26)	5.11 (3.06–8.54)	6.22	<0.001
Severity of infection Day 0						
Not in sepsis	15 (24.2)	85 (72.0)	0	1		
Sepsis	31 (50.0)	25 (21.2)	1.56 (0.33)	4.77 (2.57–8,87)	4.95	<0.001
Septic shock	16 (25.8)	8 (6.8)	1.80 (0.37)	6.01 (3.00–12.21)	4.96	<0.001
CRP/Albumin ratio Day 0 ^d^						
≤2	8 (40.0)	16 (28.1)	0	1		
>2	12 (60.0)	41 (71.9)	−0.11	0.89 (0.43–1.84)	−0.31	0.756
White cell count (×109/L) on intervention day						
≤11	28 (45.2)	62 (52.5)	0	1		
>11	34 (54.8)	56 (47.5)	1.17 (0.26)	3.21 (1.94–5.31)	4.53	<0.001
SOFA score on AMS intervention day						
≤4	25 (73.9)	108 (91.5)	0	1		
>4	47 (26.1)	10 (8.5)	1.96 (0.27)	7.10 (4.22–11.95)	7.38	<0.001
Severity of infection on intervention day						
Not in sepsis	19 (30.7)	99 (83.9)	0	1		
Sepsis	30 (48.3)	17 (14.4)	1.70 (0.30)	5.47 (3.06–9.75)	5.75	<0.001
Septic shock	13 (20.0)	2 (1.69)	2.13 (0.37)	8.44 (4.12–17.29)	5.83	<0.001

* IQR: Interquartile range; ^a^ Multidrug-resistant isolates were those producing ESBLs or *AmpC*, or carbapenem-resistant; ^b^ 53.9% missing values (*n* = 97), ^c^ 3.3% missing value (*n* = 6); ^d^ 57.2% missing values (*n* = 103).

**Table 6 antibiotics-11-00022-t006:** Pressure sore and device related factor of all-cause 30-days mortality in patients suspected with bacterial infection on extended or restricted antibiotic using simple Cox proportional hazards regression model (*n* = 180).

Variables	Event *n* = 62, Frequency (%)	Censored *n* = 118, Frequency (%)	b (SE)	Crude Hazards Ratio (95% CI)	Wald Statistic	*p*-Value
Presence of indwelling CVC						
No	26 (42.0)	96 (81.4)	0	1		
Yes	36 (58.0)	22 (18.6)	1.32 (0.26)	3.73 (2.24–6.22)	5.04	<0.001
Presence of indwelling urinary catheter						
No	16 (25.8)	73 (61.9)	0	1		
Yes	46 (74.2)	45 (38.1)	1.18 (0.29)	3.25 (1.84–5.74)	4.05	<0.001
Presence of pressure sore						
No	29 (47.5)	88 (74.6)	0	1		
Yes	32 (52.5)	30 (25.4)	0.92	2.50 (1.50–4.15)	3.53	<0.001

**Table 7 antibiotics-11-00022-t007:** Antimicrobial stewardship team related intervention on all-cause 30-days mortality in patients suspected with bacterial infection on extended or restricted antibiotic using simple Cox proportional hazards regression (*n* = 180).

Variables	Event *n* = 62 Frequency (%)	Censored *n* = 118 Frequency (%)	b (SE)	Crude Hazards Ratio (95% CI)	Wald Statistic	*p*-Value
Types of intervention						
No de-escalation	18 (29.0)	30 (25.4)	0	1		
Early de-escalation	28 (45.2)	51 (43.2)	−0.13 (0.31)	0.87 (0.48–1.60)	−0.43	0.670
Late de-escalation	16 (25.8)	37 (31.4)	−0.31 (0.35)	0.73 (0.37–1.45)	−0.89	0.373
Types of de-escalation						
No de-escalation	18 (29.0)	30 (25.4)	0	1		
Discontinuation	23 (37.1)	45 (38.2)	0.27 (1.41)	1.32 (0.08–21.03)	0.19	0.846
Changing to narrow spectrum	19 (30.7)	38 (32.2)	1.31 (1.15)	3.70 (0.38–35.57)	1.13	0.257
Shorten duration	2 (3.2)	5 (4.2)	0.86 (1.01)	2.37 (0.32–17.26)	0.85	0.393

**Table 8 antibiotics-11-00022-t008:** Univariable and multivariable analysis of prognostic factor for 30-day all-cause mortality of patients with suspected bacterial infection on extended or restricted antibiotics.

	Univariable Analysis				Multivariable Analysis			
Variables	b (SE)	Crude Hazards Ratio (95% CI)	Wald Statistic	*p*-Value	b (SE)	Adjusted Hazards Ratio (95% CI)	Wald Statistic	*p*-Value
SOFA score on AMS team intervention day								
≤4	0	1			0	1		
>4	1.63 (0.26)	5.11 (3.06–8.54)	6.22	<0.001	1.88 (0.27)	6.61 (3.90–11.18)	7.03	<0.001
CCS								
0–2		1			0	1		
≥3	0.84 (0.26)	2.32 (1.40–3.86)	3.26	0.001	0.67 (0.26)	1.97 (1.17–3.30)	2.57	0.01

b: Regression coefficient; HR = hazard ratio; CI = confidence interval; SOFA = Sequential Organ Failure Assessment.

**Table 9 antibiotics-11-00022-t009:** Impact of antibiotic de-escalation after adjusting for SOFA score on intervention day and Charlson’s comorbidity score.

Variables	b (SE)	Adjusted Hazards Ratio (95% CI)	Wald Statistic	*p*-Value
SOFA score on AMS team intervention day				
≤4	0	1		
>4	1.93 (0.27)	6.88 (4.04–11.79)	7.11	<0.001
CCS				
0–2	0	1		
≥3	0.68 (0.27)	1.97 (1.16–3.33)	2.52	0.006
Types of intervention				
No de-escalation	0	1		
Early de-escalation	−0.40 (0.31)	0.67 (0.36–1.22)	−1.30	0.194
Late de-escalation	−0.35 (0.35)	0.70 (0.35–1.41)	−0.99	0.321

b: Regression coefficient; SE: Standard error; HR = hazard ratio; CI = confidence interval; SOFA = Sequential Organ Failure Assessment. Forced entry for primary variable of interest (Types of intervention) to adjust for SOFA score on intervention day and Charlson’s comorbidity score. Multicollinearity and interactions were not observed.

## Data Availability

Data available on request due to restrictions eg privacy or ethical. The data that support the findings of this study are available on request from the corresponding author, The H.L. The data are not publicly available as data disclosure requires permission and ethical approval from Medical Research Ethical Committee (MREC), Malaysia.

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
