# Peer review of "Impact of Extended and Restricted Antibiotic Deescalation on Mortality"

_antibiotics, 2021, doi:10.3390/antibiotics11010022_

Round 1

Reviewer 1 Report

In this retrospective cohort study, the authors aimed to compare the 30-day mortality of patients on carbapenem, vancomycin and colistin according to antibiotic de-escalation strategy (early de-escalation, late de-escalation and no de-escalation) using multivariate cox proportional hazard regression analysis. Of the 180 pateints included in the analysis, 73.3% got de-escalation. They found no significant increase in 30-day mortality among patients who had antibiotic de-escalation (both early and late) compared to patients whose antibiotics were not de-escalated. They also found factors associated with 30-day mortality were SOFA score on antimicrobial stewardship intervention day, Charlson Comorbidity score and availability of CRP level.

As the authors included in the background, antibiotic de-escalation is one of the effective strategies to combat antimicrobial resistance, but it is not always the standard practice in many settings. Several studies suggested the safety of antibiotic de-escalation in various settings and various infectious diagnoses, but still data is lacking. This study will be valuable to add some insights to the body of evidence about the safety and clinical effectiveness of de-escalation practice.

This study was well designed, and I agree with the author’s conclusion that de-escalation practice is safe. But I have some suggestions and questions mainly for variable selection and data presentation.

My specific comments are below,

  1. Method section, inclusion/exclusion criteria – I think it better to include the description of antimicrobial stewardship program at Hospital Kuala Lumpur, such as how often they review antibiotic use (and how about over the weekend?), what criteria or time frame to trigger chart review for restricted antibiotics, how to communicate recommendation to providers, etc. I also think it good to show the time gap between stewardship recommendation of de-escalation and actual time of de-escalation, if data is available.
  2. Method, statistical analysis – in sample size calculation, what was the rationale for choosing 70% effect size difference to detect? I do not expect antibiotic de-escalation has such a big impact on mortality.
  3. Results – were 180 patients selected based on the sample size calculation? Or included all patients who met the inclusion criteria? Please clarify.
  4. Results – I think it better to include a table describing characteristics of patients in each group (early de-escalation, late de-escalation and no de-escalation) to help understand if there is any imbalance in characteristics.
  5. Results, table 1(c) – I wonder if the variable “availability of CRP level” is a valid one to use. What was the rationale to include this as a patient-related factor? Variable used in multivariate model should be a potential confounder. Availability of CRP level to a provider might affect de-escalation practice (exposure) but it should not directly affect patient’s outcome.
  6. Results, Lines 268-272 – it seems unrelated to the result (instruction for discussion?). Please remove.
  7. Discussion , Lines 316-317. There were unnecessary – in literature and establishing.
  8. Discussion, Lines 397-402. Related to my previous comment 5, I disagree to the author’s discussion that recommends regular and sequential CRP monitoring. In addition to my opinion that this variable might not have been appropriate to use in the model, authors did not provide information about how CRP was used when it was measured, so we do not know how it affected patients’ outcome. For those reasons, I do not think it appropriate to recommend this practice here. Would recommend to remove this part.

Author Response

Thank you for your time in reviewing this manuscript. The comments were very constructive and enlightening. Below are changes to the manuscript, and the listed changes are highlighted in yellow in the attached document. 

My specific comments are below,

  1. Method section, inclusion/exclusion criteria – I think it better to include the description of antimicrobial stewardship program at Hospital Kuala Lumpur, such as how often they review antibiotic use (and how about over the weekend?), what criteria or time frame to trigger chart review for restricted antibiotics, how to communicate recommendation to providers, etc. I also think it good to show the time gap between stewardship recommendation of de-escalation and actual time of de-escalation, if data is available. (Line 99-107) 
  2. Method, statistical analysis – in sample size calculation, what was the rationale for choosing 70% effect size difference to detect? I do not expect antibiotic de-escalation has such a big impact on mortality. (Line 173-177)
  3. Results – were 180 patients selected based on the sample size calculation? Or included all patients who met the inclusion criteria? Please clarify. (Line 182-184)
  4. Results – I think it better to include a table describing characteristics of patients in each group (early de-escalation, late de-escalation and no de-escalation) to help understand if there is any imbalance in characteristics.(Table 1 a,1b and 1c)
  5. Results, table 1(c) – I wonder if the variable “availability of CRP level” is a valid one to use. What was the rationale to include this as a patient-related factor? Variable used in multivariate model should be a potential confounder. Availability of CRP level to a provider might affect de-escalation practice (exposure) but it should not directly affect patient’s outcome. 

    Unavailability of CRP before, after, or both were also found to be associated with all-cause 30-day mortality, which highlights the importance of using sequential CRP levels. Till date no study has shown that availability of CRP levels affects mortality outcomes in bacterial infection. However, this study found that it is possible unavailability of a sequential CRP reading could have affected the choice of sequential antibiotics and hence the clinical outcome of an infection. The authors of this study felt it is important to include this variable in the analysis because in a resource limited setting where rapid diagnostic test for infection is unavailable, providers can only rely on limited laboratory data such as white cell counts which can be unreliable in immune suppressed and elderly patients, while the microbiological lab test can yield a false negative if not taken culture and sensitivity were not taken correctly. CRP levels can provide valuable information of patient’s response to an antibiotic and guide providers on the choice of antibiotic, which could possibly have confounded patient outcome indirectly. 

  6. Results, Lines 268-272 – it seems unrelated to the result (instruction for discussion?). Please remove. removed
  7. Discussion , Lines 316-317. There were unnecessary – in literature and establishing. removed
  8. Discussion, Lines 397-402. Related to my previous comment 5, I disagree to the author’s discussion that recommends regular and sequential CRP monitoring. In addition to my opinion that this variable might not have been appropriate to use in the model, authors did not provide information about how CRP was used when it was measured, so we do not know how it affected patients’ outcome. For those reasons, I do not think it appropriate to recommend this practice here. Would recommend to remove this part.

Reviewer 2 Report

Reviewer’s comments

Lin and colleagues reported that de-escalation of restricted or extended antibiotics does not significantly affect mortality of patients with or without a microbiological diagnosis. They designed a retrospective study and analyzed 180 patients, including 62 deaths (34.4%) and 118 censored events in survival analysis. Despite the importance of an antimicrobial stewardship intervention, the number of samples and duration of survival analysis (30-day survival curves) were insufficient to distinguish the impact of antibiotic de-escalation on patients between early and late interventions. Many variables shown in Tables 1 to 4 (even more details can be shown in Supplements) were not eligible for testing of infection and antibiotic resistance, before and after intervention, as the authors suggested that the mortality of patients strongly associated with a high score of Sequential Organ Function Assessment (SOFA), Charlson’s comorbidity, and the unavailability of C-reactive protein (CRP) levels. Generally, the C-reactive protein (CRP) level and erythrocyte sedimentation rate (ESR) are available to monitoring through the intervention. Although this defect in the retrospective study is vulnerable and defenseless, it may be possible for them to revise the article by using the measures of the Charlson’s comorbidity index groups to evaluate whether there is a significant difference in survival between the groups or determine the “invasive mechanical ventilation” as it is the CM score >= 3 in Table 1(a).

Author Response

We thank the reviewers for their thoughtful comments and efforts towards improving our manuscript. In the following, we highlight general concerns of reviewers that were common and our effort to address these concerns.

We would like to address comments for the second reviewer as below:  

“Despite the importance of an antimicrobial stewardship intervention, the number of samples and duration of survival analysis (30-day survival curves) were insufficient to distinguish the impact of antibiotic de-escalation on patients between early and late interventions. Many variables shown in Tables 1 to 4 (even more details can be shown in Supplements) were not eligible for testing of infection and antibiotic resistance, before and after intervention, as the authors suggested that the mortality of patients strongly associated with a high score of Sequential Organ Function Assessment (SOFA), Charlson’s comorbidity, and the unavailability of C-reactive protein (CRP) levels.”

Cut off point 30 days is to ensure that any event can be attributed to intervention and not secular trends. Agreed that the number of sample are small considering the large number of variables being tested. However, authors of this study feels that it is important to include all the listed variables as it has been shown in other studies to affect the outcome of an infection. The main reason for the large number of variables is because this study has included infections of various source types (eg. Intrabdominal, respiratory, blood etc) and the outcomes of different types of infection are affected by different factors. For instance mechanical ventilation predicts the outcome of pneumonia infection while catheter removal affects outcome of blood infection. Despite the general rule of thumb of 10 events per regression coefficients as recommended by Peduzzi et al.(1996) when handling univariate logistic regression as compared to Cox regression for a reasonably stable regression coefficients estimates, one study recommends the easing of such rule as some studies require control of confounding where more covariates than the rule of 10 or more event per variables allow (Greenland, 1989).Furthermore, Vittinghoff and McCulloch(2006) suggested the bias of coefficient estimate is mild with 5 to 9 events per variable, which this study currently stands.

“Generally, the C-reactive protein (CRP) level and erythrocyte sedimentation rate (ESR) are available to monitoring through the intervention.”  

This study had initially sought to control CRP level as potential confounding affecting 30-day all-cause mortality .This is because the relationship between rate of reduction of CRP levels with clinical outcome and prognosis have been established in various studies (Devran et al., 2012; Gradel et al., 2011; Lisboa and Rello, 2006; Moreno et al., 2010; Povoa et al., 2011). However, more than half of the cases did not have CRP levels taken before initiation of antibiotics, the effect of CRP level reduction could not be analysed, hence was dropped in multivariable analysis, and was changed to sought the association of unavailability of CRP levels with mortality. The strong association of availability of CRP levels to mortality was most likely an indirect effect, as sequential CRP levels would be a useful guide for clinicians on the effectiveness of ongoing antibiotic, especially in the immunosuppressed patients where white cell count was unreliable (Liu et al.,2008). Hence without the available CRP trends, clinicians are unable to deduce if antibiotic was effective.

Till date no study has shown that availability of CRP levels affects mortality outcomes in bacterial infection. However, this study found that it is possible unavailability of a sequential CRP reading could have affected the choice of sequential antibiotics and hence the clinical outcome of an infection. The authors of this study felt it is important to include this variable in the analysis because in a resource limited setting where rapid diagnostic test for infection is unavailable, providers can only rely on limited laboratory data such as white cell counts which can be unreliable in immune suppressed and elderly patients, while the microbiological lab test can yield a false negative if not taken culture and sensitivity were not taken correctly. CRP levels can provide valuable information of patient’s response to an antibiotic and guide providers on the choice of antibiotic, which could possibly have confounded patient outcome indirectly.

“Although this defect in the retrospective study is vulnerable and defenseless, it may be possible for them to revise the article by using the measures of the Charlson’s comorbidity index groups to evaluate whether there is a significant difference in survival between the groups or determine the “invasive mechanical ventilation” as it is the CM score >= 3 in Table 1(a).”

Charlson’s comorbidity index was evaluate and found to  demonstrated that patients with CCS ⩾3 is associated with 2-fold (95% CI 1.56,3.35; P=0.009) increase in the hazard of dying compared to those <2.

Round 2

Reviewer 1 Report

In this revised version of manuscript, the authors made modifications to the tables to make it easier to compare between groups. While I appreciate that change but I still have a concern for the use of CRP level. The authors defended the use because it might affect the choice of antibiotic if rapid diagnostic test in unavailable. I argue with following two points –

1) they are trying to show the relationship of antibiotic de-escalation (exposure) and 30-day mortality (outcome). To be a confounder, availability of CRP level needs to be associated with both exposure and outcome. If the author’s opinion is correct, availability of CRP level may affect antibiotic choice (exposure) and then mortality (outcome). In that case, they cannot say CRP level can be a confounder of the relationship between this exposure and outcome. (because CRP level does not affect outcome without involving exposure)

2) I disagree with the author’s opinion that provider needs to rely on WBC or microbiology results to judge patient is responding to treatment if CRP level is not available. More important factor would be clinical response, which should be available even in resource-limited settings. In fact, most of uncomplicated bacterial infections do not need follow-up labs.

For those two reasons, I recommend not to use the availability of CRP level as a variable in multivariable analysis. I think "unavailability of CRP level" was just a surrogate marker for “poorer clinical care”. I am afraid readers may get a wrong impression they need to utilize CRP level for follow-up of infection, which has not been proven true. I think even if that variable was completely removed, this manuscript is good to be published with the rest of results.

As a minor point, two of my previous comments were still not corrected in the revised manuscript. Please double check them. (1. Results, Lines 268-272 – it seems unrelated to the result (instruction for discussion?). Please remove. Removed 2. Discussion , Lines 316-317. There were unnecessary – in literature and establishing. Removed)

Author Response

In this revised version of manuscript, the authors made modifications to the tables to make it easier to compare between groups. While I appreciate that change but I still have a concern for the use of CRP level. The authors defended the use because it might affect the choice of antibiotic if rapid diagnostic test in unavailable. I argue with following two points –

1) they are trying to show the relationship of antibiotic de-escalation (exposure) and 30-day mortality (outcome). To be a confounder, availability of CRP level needs to be associated with both exposure and outcome. If the author’s opinion is correct, availability of CRP level may affect antibiotic choice (exposure) and then mortality (outcome). In that case, they cannot say CRP level can be a confounder of the relationship between this exposure and outcome. (because CRP level does not affect outcome without involving exposure)

2) I disagree with the author’s opinion that provider needs to rely on WBC or microbiology results to judge patient is responding to treatment if CRP level is not available. More important factor would be clinical response, which should be available even in resource-limited settings. In fact, most of uncomplicated bacterial infections do not need follow-up labs.

For those two reasons, I recommend not to use the availability of CRP level as a variable in multivariable analysis. I think "unavailability of CRP level" was just a surrogate marker for “poorer clinical care”. I am afraid readers may get a wrong impression they need to utilize CRP level for follow-up of infection, which has not been proven true. I think even if that variable was completely removed, this manuscript is good to be published with the rest of results.

Response: The variable availability of CRP was removed entirely. Hence a final model and its associated adjusted hazard ratio was amended.

As a minor point, two of my previous comments were still not corrected in the revised manuscript. Please double check them. (1. Results, Lines 268-272 – it seems unrelated to the result (instruction for discussion?). Please remove. Removed 2. Discussion , Lines 316-317. There were unnecessary – in literature and establishing. Removed)

Response: The lines 268 and 272, lines 316 and 317 was removed in the last revision. However, I suspect it maybe the running of line numbers due to multiple correction from my part.  I would like to seek reviewer’s help to restate the line numbers according to latest revision so i can make the corrections accordingly.

Reviewer 2 Report

Reviewer comments,

Lin et al. revised the manuscript by categorization of no deescalated, early-, and late-deescalated patients in Table 1, in which the frequencies of WBC counts show significant differences (P < 0.05), but with no note for the P-values. It is difficult for me to understand why the frequency of normal WBC counts (less than 11 billions per L) on day 0 and intervention was higher in no-deescalated group than in early- and late-deescalated groups.  There were many "no growth" in aetiology.  More data analysis other than the antibiotic deescalation is needed to present the statistical analysis that results in a significant difference between groups. The other data can be moved into the supplementary materials. There are many verb tense errors and plural errors to be corrected.    

Author Response

Lin et al. revised the manuscript by categorization of no deescalated, early-, and late-deescalated patients in Table 1, in which the frequencies of WBC counts show significant differences (P < 0.05), but with no note for the P-values. It is difficult for me to understand why the frequency of normal WBC counts (less than 11 billions per L) on day 0 and intervention was higher in no-deescalated group than in early- and late-deescalated groups.  There were many "no growth" in aetiology.  More data analysis other than the antibiotic deescalation is needed to present the statistical analysis that results in a significant difference between groups. The other data can be moved into the supplementary materials. There are many verb tense errors and plural errors to be corrected.    

*Statistically significant difference were found between no de-escalation vs early de-escalation, and no de-escalation vs late de-escalation. (Notes inserted at the bottom of Table 1)

The frequency of normal WBC counts (less than 11 billions per L) on day 0 and intervention was significantly higher in no-deescalated group than in early- and late-deescalated groups. This was because primary team generally refuse to de-escalate once patients has shown to respond to an antibiotic regimen as shown in normalization of white cell count, and would tend to continue and complete the antibiotic regimen. (Line 252 -258)
